# The Anti-Fungal Activity of Nitropropenyl Benzodioxole (NPBD), a Redox-Thiol Oxidant and Tyrosine Phosphatase Inhibitor

**DOI:** 10.3390/antibiotics11091188

**Published:** 2022-09-02

**Authors:** Gina Nicoletti, Kylie White

**Affiliations:** STEM College, RMIT University, Melbourne, VIC 3001, Australia

**Keywords:** nitroalkenyl benzenes, nitropropenyl benzodioxole, tyrosine phosphatase, redox-active thiols, cysteine

## Abstract

Phylogenetically diverse fungal species are an increasing cause of severe disease and mortality. Identification of new targets and development of new fungicidal drugs are required to augment the effectiveness of current chemotherapy and counter increasing resistance in pathogens. Nitroalkenyl benzene derivatives are thiol oxidants and inhibitors of cysteine-based molecules, which show broad biological activity against microorganisms. Nitropropenyl benzodioxole (NPBD), one of the most active antimicrobial derivatives, shows high activity in MIC assays for phylogenetically diverse saprophytic, commensal and parasitic fungi. NPBD was fungicidal to all species except the dermatophytic fungi, with an activity profile comparable to that of Amphotericin B and Miconazole. NPBD showed differing patterns of dynamic kill rates under different growth conditions for *Candida albicans* and *Aspergillus fumigatus* and was rapidly fungicidal for non-replicating vegetative forms and microconidia. It did not induce resistant or drug tolerant strains in major pathogens on long term exposure. A literature review highlights the complexity and interactivity of fungal tyrosine phosphate and redox signaling pathways, their differing metabolic effects in fungal species and identifies some targets for inhibition. A comparison of the metabolic activities of Amphotericin B, Miconazole and NPBD highlights the multiple cellular functions of these agents and the complementarity of many mechanisms. The activity profile of NPBD illustrates the functional diversity of fungal tyrosine phosphatases and thiol-based redox active molecules and contributes to the validation of tyrosine phosphatases and redox thiol molecules as related and complementary selective targets for antimicrobial drug development. NPBD is a selective antifungal agent with low oral toxicity which would be suitable for local treatment of skin and mucosal infections.

## 1. Introduction

Post-translational protein phosphorylation and integrated, highly coordinated networks of phosphate signaling pathways control cell growth and development and the cellular response to internal and external stimuli [1,2]. Transient, reversible oxidation by redox-active molecules regulates kinases, phosphatases, thiol-containing functional proteins and components of other interacting cellular pathways [3,4]. Microorganisms have evolved complex and functionally divergent signaling and redox mechanisms to meet the challenges of survival in diverse and rapidly changing environments [5,6].

Reversible phosphorylation of proteins on Ser, Thr and Tyr in fungi regulates the complex and integrated signaling pathways involved in transcription, the cell cycle, mating, vegetative growth and morphogenesis, maintenance of cell integrity and responses to external stress [7,8]. Pathway activation by phosphorylation of proteins, and termination by dephosphorylation, is achieved by the reciprocal actions of protein kinases and protein phosphatases and the sensing phosphoproteins mediating downstream cellular activities. Kinases regulate the initiation and amplitude of phosphate signaling. Phosphatases regulate the rate and duration of signaling by dephosphorylating kinase autophosphorylation or tyrosine regulatory sites, or downstream phosphorylated proteins.

Microscopic fungi show lower conservation and greater diversity in phosphoproteins compared to higher eukaryotes, with the functionally important phosphoproteins being the more highly conserved [6,7,8,9]. Kinase and phosphatase orthologs are heterogeneously distributed across fungal species and often show diversified functions depending on habitats and activities [10,11,12,13]. Orthologs of protein tyrosine kinases of higher eukaryotes are lacking in unicellular fungi [14]. The main burden of regulating growth, developmental and stress resistance in fungi is undertaken by the serine/threonine kinases (STK), cyclin-dependent kinases (CDK) and the mitogen-activated protein (MAP) kinases and their associated phosphatases.

MAP kinases are highly conserved signaling molecules that act in modules which share components, enabling pathway interactions [15,16]. The ‘effector’ kinase in the ‘MAPK’ module, is a dual specificity Tyr/Thr kinase, activated by phosphorylation on catalytic Tyr and Thr residues by an upstream STK MAP kinase. The module ‘MAPK’ activates downstream kinases, phosphatases and tyrosine phosphoproteins (PTyr), resulting in the required signal responses. MAPK can be negatively regulated by serine, threonine and tyrosine phosphatases and dual specificity MAPK-specific phosphatases (MAPKP) that dephosphorylate either or both PTyr and PThr catalytic residues [17]. The Mpk1 ‘cell wall integrity’ pathway, the ‘high osmolality glycerol’ (Hog1) stress response pathway and the Fus3/Kss1 mating and filamentation pathways, independently and cooperatively, play essential roles in fungal growth, development, virulence and adaptation to stress [15,16,18]. Fungal MAP kinases are generally highly conserved but show considerable functional divergence in number of MAPK genes and in activation loop motifs [6,11,19].

Phosphatases important in fungal growth and pathogenicity comprise the major Ser/Thr phosphatases, and cysteine-based tyrosine phosphatases (PTP) which share an essential cysteine catalytic motif. Tyrosine phosphatases have evolved in structurally and functionally diverse families, are more numerous, less conserved and with lower substrate specificity than their cognate kinases. Structural and sequence differences in the catalytic loop govern substrate specificity and enzyme function. PTP include tyrosine-specific (sPTP), low molecular weight PTP (LMWPTP) and dual-specificity phosphatases (DSP) capable of dephosphorylating Tyr, Ser and Thr. DSP have 3 catalytic motifs, a shallow catalytic site to accommodate phosphotyrosine, phosphoserine and phosphothreonine. They are more functionally diverse than sPTP, affect a wider variety of substrates and play a major role in tyrosine phosphate signaling in fungi [2,7,9]. PTP are regulated by reversible oxidation of the catalytic cysteine and undergo transient inactivation by low-level oxidants from normal metabolism and reactivation by cell reductants, enabling efficient enzyme recycling [20].

Cysteine has many functional and regulatory roles in cells arising from the redox activity of the electron-rich, nucleophilic sulfhydral group. Sulfur is highly susceptible to sequential, reversible and irreversible oxidation by redox-active oxygen, nitrogen, metal and electrophilic species generated from normal metabolism or from cell responses to external stress stimuli [21,22]. The strong redox activity of cysteine enables it to have both functional and regulatory roles as a catalytic nucleophile in many enzymes, particularly PTP and the oxidoreductases, and in LMW thiols which regulate the maintenance of cytosol redox homeostasis [21,23].

Many physiological processes in cells are rapidly altered in response to changes in cellular redox conditions. The cysteine SH group functions as a major sensor and regulator of redox-active molecules by cycling between the oxidized and reduced states. Transient reversible oxidation of the sulfhydral group to sulfenic acid and formation of a disulfide bridge is the most common redox modification in bacteria interacting with oxidant or reductant molecules. Cysteine-based molecular switches act as major binary on/off switches in the regulation of enzyme function, the activation of transcription factors and the structure and function of cysteine-containing proteins [24,25].

Fungi are continuously exposed to changing oxidative and chemical stresses. Complex antioxidant enzyme systems, redox sensors, LMW redox buffers and thiol-containing proteins, maintain cells in a chemically reduced state. Redox species and antioxidant enzyme systems differ across phylogenetic groups and play diverse roles in growth and differentiation, parasitism, plant pathogenicity and rapid responses to oxidative stress [26,27,28,29]. In fungi, the cysteine-containing tripeptide, glutathione is the major small molecule redox buffer along with free cysteine and redox-active thiol residues in proteins and low levels of free cysteine. Fungal antioxidant enzyme families, include the cysteine-based glutaredoxin, and the thioredoxin and peroxiredoxin systems [27,30].

Nitroalkene benzenes (NAB) are strong electrophiles and oxidants. The nitroethenyl and nitropropenyl substituents are acceptors of Michael addition to the electrophilic carbonyl group from nucleophilic cysteine thiolate and other thiols [31,32,33]. NABs can thus potentially target the thiolate group in cysteine-based enzymes, redox-active thiol-containing molecules in anti-oxidative defence systems and redox proteins in metabolic pathways [34,35,36,37].

The nitroethenyl and nitropropenyl moieties of NAB are responsible for their antibacterial and antifungal activity [38,39]. In human cells NAB derivatives have been shown to inhibit human telomerase [40], platelet aggregation and SH2 domain Tyr phosphorylation-activated kinases Src and Syk [41], induce apoptosis in rat tumour cells [42] and inhibit human tumour cell lines [43]. NAB derivatives reversibly inhibit the enzymic activity of PTP1B, Shp1, bacterial YopH and CDC45 by oxidation of the catalytic cysteinyl residue. Enzymic activity is increased with increasing electrophilicity and reversed in the presence of mercaptoethanol [38,44,45,46,47,48] (Appendix A). Substituents on the benzene ring modify enzymatic activity, antimicrobial and anti-tumour activity [38,49,50]. Nitropropenyl benzenes have greater antibacterial and antifungal activity than the corresponding nitroethenyl compounds [38,44,49]. Antibacterial activity positively correlates with the greater negative redox potential of the propenyl compared with ethenyl side chain, but not with lipophilicity [49].

Nitropropenyl benzodioxole (NPBD) (Figure 1, Appendix A) is the most broadly active of 30 NAB derivatives [38,50]. NPBD is a lipophilic, cell permeable, neutral tyrosine mimetic which competitively inhibits the enzymic activity of tyrosine phosphatases [39]. NPBD inhibits the enzymic activity of PTPB1 and YOP and to a lesser extent CDC45 [47,49]. It binds to Cys215 in the catalytic site of PTP1B [44]. NPBD antibacterial activity against representative bacterial species, measured by MIC and MBC, is competitively reversed in the presence of dithiothreitol and to a lesser extent, excess cysteine [39].

NPBD activity against a broad range of bacterial species shows considerable variation, reflecting the diversity of cysteine-based enzymes and redox active thiol molecules across bacterial species and indicative of selectivity of action [39]. NPBD antibacterial activity is significantly reversed in the presence of dithiothreitol and to a lesser extent, excess cysteine [39]. The heterogeneity of distribution, structure and function of cysteine-based enzymes, redox-thiol signaling molecules and cysteine-containing proteins indicates that possible NPBD targets would vary in type, number and susceptibility across fungal species. Such targets are in constant flux, produced transiently, in low concentrations and the identification of specific targets in susceptible species would therefore require complex investigations. Few cysteine-based enzymes and redox-active molecules have been characterised in fungi [26,27,51,52]. Involvement of such targets in physiological functions is usually identified by modification or loss of a physiological effect in null mutants or inhibition by inhibitors, supported by many references in this paper.

Severe, invasive, often intractable mycoses are an increasing cause of disease and mortality, the fungal pathogens varying phylogenetically and in global geographic distribution [53]. Chemotherapy for severe infections, largely based on azoles, echinocandins and polyenes, is limited in availability, of varying therapeutic success and has high mortality rates [54]. Identification of new targets and development of new fungicidal drugs is required to augment the effectiveness of current chemotherapy, resistance in major pathogens and drug toxicity.

We report here on phenotypic effects of NPBD across phylogenetically diverse saprophytic, commensal and parasitic fungal species. The data illustrate the heterogeneous distribution of cysteine-based enzymes and redox-active thiol molecules, identify pathogens of interest and useful attributes for an antifungal drug candidate. NPBD in vitro activity is evaluated against Amphotericin B (AMB) and Miconazole (MCZ) which have a similar antifungal spectrum. The shared metabolic activities of NPBD, AMB and MCZ are compared by a brief literature review. The proposal that PTP and redox thiols can be selective targets in fungi is supported by a literature review of tyrosine phosphate and redox signaling in fungal species. The data presented supports the proposal that NPBD would be a suitable candidate for development as an agent to treat mucocutaneous opportunistic fungal infections.

## 2. Results and Discussion

### 2.1. NPBD Shows Broad-Spectrum, Rapid Fungicidal Activity

The activity of NPBD, AMB and MCZ was measured by Minimum Inhibitory Concentration (MIC) assay using vegetative cell inocula and performed concurrently in the same laboratory, permitting an internally controlled evaluation of comparative activity. Reproducibility across assays was high, indicated by low standard deviations. Hyphae may give higher MIC and Minimum Fungicidal Concentration (MFC) titres than microconidial inocula, the differences varying with drug and species [55,56]. The AMB MIC (Table 1) are within, or close to CLSI Performance Standards MIC ranges (mg/L) for type strains of *A. flavus* (0.5–8), *A. fumigatus* (0.5–4), *S. apiospermum* (1–16), *T. rubrum* (0.5–2) and *Candida parapsilosis* (0.5–4) [57].

NPBD showed broad, strong and relatively uniform antifungal activity across 27 saprophytic, commensal and parasitic species from 3 orders and 12 families (Table 1). Spectrum and activity levels were comparable to those of AMB and MCZ.

All agents showed high fungicidal activity for two significant pathogens, *C. neoformans* and *S. apiospermum* (Table 1). NPBD was ≤2-fold less active than AMB and MCZ for saprophytic filamentous species, but overall, more rapidly fungicidal, evidenced by its lower ratio of fungal group MFC/MIC, a static index of multiples of the MIC required for lethality at one time point (Table 2). NPBD was more inhibitory but considerably less fungicidal than MCZ to the parasitic dermatophytes. It was less active against Candida species than AMB and MCZ but more rapidly fungicidal measured by the MFC/MIC ratio (Table 2). NPBD is comparably active against the hyphal forms of thermally dimorphic species of *Fonsecaea*, *Hortaea*, *Phialophora*, *S. apiospermun* and *C. neoformans* (Table 1, Table 2). NPBD has also been reported to be active against *Blastomyces dermatidis*, *Histoplasma capsulatum* and *Coccidioides* species (MIC_90_ 0.25–2 mg/L), *Cryptococcus gatii* (2 mg/L), and *Candida glabrata* (0.5–2 mg/L) (Appendix A). NPBD was also highly active against *Pneumocystis jirovecii* (*carinii*) and *Pneumocystis murina*, IC_50_ on day 3 of exposure was <0.1 and 0.174 mg/L, respectively [60].

The fungicidal activity pattern shows NPBD is acting on metabolic functions vital to the growth and survival of saprophytic and commensal species. Variation in activity between and within related fungal groups illustrates the functional diversity of PTP and redox-active molecules. The physiological outcomes reported reflect the balance of positive and negative metabolic effects on the functions of those cysteine-based enzymes and redox-active thiol molecules that are susceptible to interference by NPBD. The target molecules are transiently present, in low concentrations making identification of specific targets and substrates technically difficult and requiring investigation of suspect molecules in each species of interest [61,62]. The identification of enzymic or redox targets for NPBD in the assayed species is beyond the scope of this paper. A brief review of characterised PTP and redox active molecules may indicate possible targets for NPBD.

Progression and check-point control in the eukaryote cell cycle are regulated by highly conserved CDK and counteracting MAPKP. The DSPs Cdc25 and Cdc14 activate specific CDK by removal of inhibitory tyrosine and threonine phosphorylation and are important regulators of cell cycle progression [63].

Cdc25 isoforms are highly conserved in fungi and are crucial in maintaining the activity of CDK, allowing cycle progression. Down-regulation of Cdc25 in response to DNA damage promotes cycle arrest at mitotic entry to allow DNA repair. Continued suppression of Cdc25 results in cell arrest and progress to apoptosis [64]. Cdc25 phosphatases are highly susceptible to rapid oxidation of the catalytic cysteine and are protected from continuing and irreversible inactivation by formation of a disulfide bond with an adjacent allosteric cysteine and its reduction by thioredoxin/thioredoxin reductase [65]. Naphthoquinones are electrophiles and oxidants which show antifungal activity [66]. Naphthoquinones oxidise Cdc25 causing mitotic arrest and cell death which is reversed by dithiothreitol and cause tumour regression in a zebrafish xenograft model of colorectal cancer cells over-expressing Cdc25 [67]. Cdc25 isoforms are credible targets for NPBD inactivation by dephosphorylation and oxidation.

Cdc14 is highly but not uniformly conserved across fungi and higher eukaryotes and shows specific and differing functions. In *S. cerevisiae* and *C. albicans*. Cdc14 is required for reversal of CDK phosphorylation and mitotic exit, but not in *Schizosaccharomyces pombe* and some higher eukaryotes [66]. Cdc14 in *C. albicans*, is required for inhibition of cell separation and the transition to hyphal growth [68]. Cdc14 substrates are variously involved, positively and negatively, in rDNA transcription, DNA replication, cytokinesis and cell separation, DNA damage response and DNA repair and morphogenesis [66,68,69,70]. The activity and substrate specificity of Cdc14 homologs in plant pathogenic fungi are highly conserved and their deletion impairs virulence [71,72]. In *Aspergillus flavus*, Cdc14 positively regulates growth, development, plant infectivity and resistance to osmotic stress but suppresses aflatoxin production [72]. NPBD interference with Cdc14 activity would depend on the functions of its many regulated proteins but is likely to be detrimental overall. Most specific functions of fungal phosphatases reported are associated with dephosphorylation of serine and threonine with very few to date directly attributed to tyrosine dephosphorylation [63,69].

The cell wall integrity (Mpk1), stress regulating (Hog1) and Fus3/Kss1 mating and filamentation pathways play interactive roles in cell cycle progression, cell wall biosynthesis, cell differentiation, vegetative and invasive growth, stress adaptation and virulence in fungi [73,74,75]. These pathways are activated by diverse stimuli and are partly and variously negatively regulated by cognate sPTP and MAPKP. Phosphatase orthologues are widely distributed and show structural and functional divergence and differing pathway interactions [15,76]. Orthologues are identified in *A. fumigatus* [77], *C. albicans* [78], *C. neoformans* [79], and *F. graminearum* [10].

The Mpk1 pathway monitors cell wall integrity (CWI), promotes wall biosynthesis and some stress responses and is well conserved with some functional and regulation differentiation [74]. The Kss1/Cek1 pathway in *C. albicans* is involved in cell wall construction, chlamydospore formation, filamentation and invasive growth. Cek1 is negatively regulated by Ptp2, Ptp3 and DSP Cpp1, the latter being positively regulated by Hog1 [79,80]. In *C. neoformans* Ptp2 dephosphorylation activates Hog1 to promote a stress response. Ptp2 is essential for vegetative growth. Ptp1 plays minor supporting roles in stress response and growth [76,79]. Suppression of sPTP and DSP MAPKPs which are negative regulators of Mpk1 and Kss1 pathways, could enhance vegetative growth or, conversely, prolong kinase activation with possible deleterious effects on cell homeostasis.

Hog1, a key regulator of cell responses to external stress, has diverse functions and regulatory mechanisms in fungi. For most fungal species Hog1 is activated by tyrosine phosphorylation and deactivated by negatively regulating sPTP and MAPKP. Adaptation to oxidative stress depends on coordinated positive and negative interactions between Hog1 and other MAPK pathways resulting in Hog1 involvement in growth, development and virulence [76,81,82]. Hog1 shares components with the CWI and Fus3/Kss1 pathways and, when activated, to permit the stress responses, downregulates Mpk1 and Kss1, affecting cell cycle progression, cell wall construction, growth and morphogenesis [81]. In *C. albicans* Cpp1 mediates interaction between the Hog1 and Cek1 pathways, Cpp1 expression being positively regulated by Hog1 and contributing to suppression of Cek1 [83]. Inhibition of sPTP and MAPKP negatively regulating Hog1 could result in continuing activation leading to prolonged cycle arrest and down regulation of MAPK pathways, with deleterious effects [80,81]. Hog1 in *C. neoformans* is phosphorylated under normal conditions and activated by Ptp2 and Ptp1 dephosphorylation. Hog1 regulates growth, development and virulence attributes in *C. neoformans* and positively regulates the expression of Ptp1 and Ptp2, its negative feedback regulators ([84]. In *C. neoformans* Ptp2 overexpression suppresses the hyperactivation of Hog1 and suppression of Ptp2 is lethal [79]. For many species, inhibition of tyrosine phosphatases down regulating Hog1 could result in prolonged activation of Hog1 and inactivation of related MAPK pathways, with deleterious effects [72,83,85]. Fludioxonil, a benzodioxole with an electrophilic carbonitrile-containing substituent, causes lethal hyperactivation of Hog1 [86]. Overexpression Ptp2 increased, and Ptp2 deletion decreased, resistance to fludioxonil in plant pathogen *Magnaporthe oryzae* [87]. Other than the well-reported highly conserved primary target(s), the broader metabolic effects of antifungal agents have not been well investigated. Drug-specific resistant strains which are originally categorised by a ‘specific target’ assay, can show unique metabolic changes which are indicative of a diversity of drug-induced metabolic effects. Such changes may increase or decrease resistance to the originating drug class. The polyene macrolides and several azole drugs have complex and diverse metabolic effects on fungi [51,88,89,90,91,92,93].

High levels of redox-reactive species arrest cell growth in fungi resulting in expression of anti-oxidative genes and a switch to cell differentiation [30]. Oxidation of LMW thiol buffers, redox-active cysteine residues in proteins and cysteine-based enzymes will interfere with redox homeostasis and contribute to NPBD lethality. Addition of excess dithiothreitol to culture media significantly inhibited the antibacterial activity of NPBD, confirming the contribution of thiol oxidation to bactericidal activity [39].

MAPK tyrosine phosphatases are likely targets for NPBD inhibition. Negative regulation of MAPKs, which respond to diverse stimuli and act in integrated pathways, is critical to prevent prolonged activation of affected kinases which would result in deleterious effects. Dual specificity and sPTP which positively regulate aspects of growth, survival or stress responses are likely direct targets. NPBD direct oxidation of cysteine-based redox active molecules and redox thiols will contribute to growth inhibition and cell death.

### 2.2. NPBD Kills Replicating and Non-Replicating C. albicans Blastospores and A. fumigatus Hyphae and Microconidia

Fungal populations can be heterogeneous and contain subpopulations exhibiting phenotypic variability, including drug tolerance. Drug tolerant cells mobilise stress responses, allowing non-target-directed resistance to the drug and time-dependent population growth above the MIC, but retain inherent drug susceptibility when the drug stress is removed [94]. Investigation of the dynamic fungicidal effect of agents under different growth conditions assist in identifying interactions between agent and microorganism, characterise fungicidal potential and detect drug response heterogeneity in a population. Kill rates can be investigated by time-kill assays which quantify over time the effect of agent concentrations on viability and replication, assessed by the ability to form colonies and expressed as time-kill (TK) curves and log_10_ population reduction factors (RF) [95].

We estimated the rate of kill of replicating blastospores of *C. albicans* in broth under optimal conditions and of non-replicating blastospores of *C. albicans* and hyphae and microconidia of *A. fumigatus* in water at ~20 °C. Under optimal growth conditions, NPBD (0.5× to 4× MICi) showed strong dose-dependent population reduction for blastospores of *C. albicans*, 4× MICi reducing cell counts by >5 log_10_ at 4 h (Figure 2a). For non-replicating blastospores in DW the kill rate was slower and dose-independent, NPBD 8× and 16× MIC being equi-effective (RF > 5 at 6 h) (Figure 2b). NPBD (16× and 32× MICi), reduced the viability of non-growing hyphal fragments of *A. fumigatus*, RF > 4 at 48 h (Figure 2c). NPBD was sporicidal for microconidia of *A. fumigatus* in water. NPBD at 16× MICi produced a >3 log_10_ reduction of spores able to germinate for exposures > 6 to 24 h (Figure 2d). The MFC/MIC ratios of NPBD under optimal growth conditions for *A. fumigatus* hyphae (2.6) and for *C. albicans* blastospores (1.1) are lower than for AMB (8 and 8 respectively) demonstrating a more rapid kill rate for NPBD.

AMB showed greatly varying MFC titres and kill patterns across 7 *Candida* species. with the greatest kill rate for *C. albicans*, 8× and 16× MIC showing >4 log_10_ reduction at 4 h and 6 h using a lower inoculum density 10^5^ cfu/mL [96]. Another study with *C. albicans* using an inoculum density of 10^6^ cfu/mL exposed to 4× and 16× MIC reported a 2 and 3 log reduction, respectively, at 8 h [97]. *C. albicans* cells exposed to AMB at 13× MICi, show >3 log_10_ colony count reduction at 10 h and changing proportions of non-metabolising, replication-incompetent cells with time [98]. AMB has demonstrated a dose-independent kill rate for optimally growing *A. fumigatus* hyphae causing a 2 log_10_ reduction at 48 h at 4× and 16× MIC [99]. Comparison of these reported results with results above, despite assay variations, supports a faster and earlier fungicidal rate of NPBD for replicating and non-replicating vegetative cells of *A. fumigatus* and *C. albicans*.

*A. fumigatus* conidia germinate into short hyphae by 6–8 h at 37 °C in vitro. In lung tissue of neutropenic mice, 80% of conidia germinate by 12–14 h post-infection [100]. Microconidia of *A. fumigatus* are highly dispersible, and germination in vivo is a key factor in pathogenicity [101]. The ability to rapidly kill microconidia at the mucosal surface would be advantageous for an antifungal drug.

The dose-independent fungicidal action of NPBD in non-replicating *C. albicans* blastospores and hyphal fragments of *A. fumigatus* in water suggests interference with PTP essential for cell viability. The more rapid killing of *C. albicans* under optimal conditions in broth at lower multiples of the MIC suggests the additional suppression of PTP involved in cycle progression and replication and the promotion of oxidative stress are important for rapid reduction of growing populations. The ability to kill dormant or slowly metabolising cells at achievable in vivo concentrations can be important for chemotherapeutic efficacy [94].

### 2.3. NPBD Did Not Induce Resistance or Tolerance in Fungal Strains on Long Term Exposure

Acquired resistance to the polyene, azole and echinocandin drugs is reported for major pathogenic genera *Candida*, *Cryptococcus*, *Aspergillus*, *Scedosporium* and *Fusarium* [94]. For some species, sub-populations show drug tolerance, dependence on stress response pathways and reversion to intrinsic susceptibility. Long term exposure of a strain to subinhibitory drug concentrations in vitro can select for drug-tolerant sub-populations or induce mutations to resistance. Resistance is indicated by an increase in MIC greater than the accepted titre variation for the susceptible strain and which is sustained in the absence of drug. Tolerance is detected by a persisting MIC above the MIC range during exposure with reversion to the susceptible MIC range in the absence of drug [95]. Continuous exposure of bacterial species to NPBD for 16 weeks did not induce resistance or drug tolerance [39].

Fungal strains were continuously exposed to NPBD for 12 weeks and changes in MIC monitored (Figure 3). After passages in drug free broth the MIC of NPBD-exposed and unexposed colonies differed ≤4-fold. No resistance or tolerance to NPBD was observed for any strain. The lack of cross resistance suggests the metabolic targets differ between the compared drugs. Moderately toxic antifungal drugs generally attack few and specific metabolic targets with little redundancy and identified resistance mechanisms generally relate to target-associated metabolic pathways or are generic responses such as drug efflux [102]. Several PTP may be involved in a physiological response to NPBD exposure and resistance would likely require multiple mutations. Inhibition of multiple PTP and other cysteine-dependent enzymes, other protein thiols and LMWT by NPBD would greatly reduce the likelihood of mutations to resistance or the emergence of persister cells.

AMB and MCZ activate oxidative stress responses in fungi and have a very low propensity to induce resistant strains [88,90,102]. AMB resistant strains showing loss of *erg2* and *erg3/11* exhibit metabolically costly stress responses which lower survival and virulence [103]. AMB and fungicidal azoles induce ROS-dependent cell death in *C. albicans* and *Cryptococcus* spp. [92]. Both promote the survival of high-level drug tolerant persister cells with activated oxidative enzymes in *Candida* species. [104]. AMB resistance in *Aspergillus terreus* is accompanied by an enhanced oxidative stress response and increased catalase and superoxide dismutase (SOD) activity [105]. Interactions between AMB and azole drugs in vitro indicate indifferent or antagonistic effects on activity [106,107,108]. 1,3,4-thiadiazole derivatives are strong oxidants with broad biological effects. A thiadiazole derivative in combination with AMB induced high oxidative stress and irreversible cell damage in fungal strains [109]. NPBD has not been tested for synergistic or antagonistic effect on AMB or MCZ. The complementarity of their function as oxidants suggests a possible synergistic interaction.

As a strong oxidant interfering with complex phosphatase and redox signaling pathways with multiple lethal downstream effects, NPBD is unlikely to give rise to resistant mutant to the same degree and the major classes of antifungal drugs.

### 2.4. Comparison of Activity Profiles and Mechanisms of NPBD, AMB and MCZ

NPBD, AMB and MCZ are multi-mechanism agents which differ in specific mechanisms but share many physiological effects in cells. All have broad antimicrobial profiles and do not induce significant resistance in key pathogens. NPBD is broadly active across phylogenetically diverse bacterial species with a much greater variation in activity than the uniform activity across all fungal species reported here [39]. This reflects the greater diversity of distribution and function of bacterial tyrosine phosphatases [110]. MCZ is active against Gram-positive bacteria [111]. AMB and MCZ are active against *Plasmodium*, *Leishmania*, *Trypanosoma* and *Acanthamoeba* [112]. NPBD inhibits *Trichomonas vaginalis* in vitro, asexual development of *Plasmodium falciparum* in human erythrocytes and *Eimeria* infections in chicks (Nicoletti, unpublished). AMB and MCZ promote oxidative stress in fungi [113,114]. All three agents inhibit human tumour cell lines. AMB is cytotoxic to breast cancer cells and potentiates doxirubicin cytotoxicity [115]. MCZ induces ROS generation and apoptosis in breast and human colon adenocarcinoma cells and in mouse adenocarcinoma xenografts [116,117]. NPBD is toxic to non-small cell lung cancer A549 cells (EC_50_ 1.7 μM) (Appendix A, [118]). Closely related NABs show broad antitumour activity. NEB inhibits osteosarcoma cell lines [119]. Hydroxy-methoxy nitropropenyl benzene (CYT-Rx20) induces reactive oxygen species (ROS) generation and apoptosis in breast cancer cells [120]. CYT-Rx20 suppresses the P13/AKT and STAT3 pathways in oesophageal tumour cells, depletes GSH and inhibits GSH reductase in A549 cells [121,122].

Cytochrome P450 monooxygenases (CYP) are haem-thiolate tetrapyrrole proteins which catalyse the multi-step oxygenation of multiple endogenous and exogenous metabolites [123,124]. CYP isoforms share a common active site bearing a haem-Fe complex which is ligated to the CYP protein via a thiolate ligand from a flanking essential cysteine residue. During the catalytic cycle, electron transfer from carriers to the activated haem complex results in oxygenation of substrates. A portion of activated oxygen is leaked from the catalytic intermediates contributing to ROS levels. Induction of CYP or suppression of cysteine-based peroxiredoxins contribute to excess ROS generation, contributing to oxidative stress [123].

CYP isoforms are functionally and structurally diverse, the size and shape around the active site and the varying position of critical cysteine residues allowing selective catalytic specificity for a vast number of substrates [123,124]. Fungal CYP, have conserved characteristic motifs but show great structural and functional diversity to enable adaptation to multiple and changing environments. They are involved in primary and secondary metabolism, detoxification and degradation of xenobiotics, and adaptation to stress [125]. Cyp51 and fungal specific CYP61, widespread in fungi are essential in membrane ergosterol biosynthesis and are inhibited by azole-based drugs [126]. MCZ inhibits fungal Cyp51 and Cyp61 and variably inhibits human Cyp1A2, Cyp2D6, Cyp2C9 and Cyp2C19 with IC_50_ from 0.33 µM to 25 µM [127,128]. NPBD inhibits human Cyp1A2 (EC_50_ 0.53 µM) and at 20 µM shows 35% inhibition of Cyp2C9, but EC_50_ >20 µM for Cyp2C19, Cyp2D6, Cyp3A4/5 (Appendix A, [129,130]). NPBD CYP inhibition is likely due to oxidation of the cysteine ligating haem Fe, thus interfering with CYP substrate binding and catalysis.

AMB, administered by IV, is used systemically for severe intractable mycoses, has poor pharmacokinetics and causes infusion-related reactions and nephrotoxicity [131]. MCZ has low oral toxicity, but systemic toxicity constrains use and MCZ is mainly used for mucocutaneous infections [132]. NAB analogues show varying toxicity patterns in zebrafish assays. NPBD, EC_50_ (0.92 µM) in zebrafish embryos reduced the heartbeat rate, spontaneous movement and impaired eye development and tail extension but caused no malformations [38]. The zebrafish toxicity of AMB (LD_50_ 0.1 µM) is of comparable level [133]. MCZ (1–100 µM) shows dose-dependent toxicity to rat neonatal cardiomyocytes, 3 µM inhibiting beating frequency [134]. MCZ (2.3–8.4 µM) blocks K+ channels in HEK cells [135]. NPBD toxicity for zebrafish was not predictive of rodent toxicity. NPBD has low oral toxicity to rodents (7 day repeat dose NOEL < 300 mg/kg) and low oral absorption, generally <2.5% (Appendix A). NPBD given at 25 mg/kg by IV bi-weekly for three weeks did not adversely affect weight or general wellness of nude athymic mice (Appendix A). NPBD binds strongly to human serum albumin and shows greatly reduced antibacterial activity in the presence of plasma [39]. Low IV toxicity is likely due to levels of free drug well below 25 mg/kg. Acute toxicity IV LD_50_ for AMB-desoxycholate in the mouse is 3 mg/kg and >40 mg/kg for lipid-based formulations [136].

NPBD shares complementary metabolic effects with AMB and MCZ that contribute to antimicrobial and anti-tumour activity.

## 3. Conclusions

NPBD is broadly and rapidly fungicidal to vegetative cells and microconidia and does not induce drug tolerant or resistant cells on long term exposure in vitro. Given its multiple and complementary effects on redox molecules, development of resistant strains in therapeutic use is unlikely. It has low oral toxicity in rodents and chickens and no adverse effects on health, wellbeing and behaviour. NPBD has an activity profile in vitro similar to that of AMB and MCZ which are systemically relatively toxic and with circumscribed therapeutic uses. NPBD meets desirable pharmacological requirements for a cell permeable, small molecule tyrosine mimetic drug for systemic use. The data presented in this report supports the suitability of NPBD for the treatment of mucocutaneous infections. Inhibition of fungi at the mucosal and epithelial surfaces, major entry points for fungal pathogens, could limit and delay progress to invasive infections. It may be useful as an adjunct drug in chemotherapy for severe, persistent or intractable systemic mycoses.

All the major current antifungal agents target highly conserved metabolic functions of actively growing fungi, such as biosynthesis of cell membrane and cell wall components, mRNA and DNA. These selective targets have given rise to resistant mutants causing therapeutic problems. This situation has given rise to consideration of alternative novel targets in fungi. There is increasing interest in inhibitors of redox-sensitive fungal targets such as CYP 450, heme metalloproteins, transcription factors and MAPK signaling components [102,137,138,139,140]. Redox targets in tumor biology are also being investigated and may be applicable as targets for antifungal agents [102,141,142]. NPBD and other nitroalkenyl derivatives can make a useful contribution to the drug target arsenal.

## 4. Materials and Methods

### 4.1. MIC and MFC Assay

Antifungal activity of NPBD, amphotericin B and miconazole were determined for species listed in Table 1 using MIC and MFC broth microdilution methods, modified for filamentous fungi by use of hyphal inocula for filamentous species so that agent activity was assayed on vegetative forms [58,59]. Log_2_ dilution sets of NPBD, amphotericin B (Sigma Aldrich, St. Louis, MO, USA) and miconazole (Sigma Aldrich) were prepared in RPMI-1640 (Sigma Aldrich) at 2× final concentrations, 0.03–128 mg/L for NPBD and MCZ and 0.03–16 mg/L for AMB. For filamentous species hyphal suspensions from overnight cultures were filtered through gauze, vortexed to fragment hyphae and standardised by haemocytometer and colony count. Final inoculum density was 5 × 10^3^–1 × 10^4^ cfu/mL, MIC for dermatophytes were read at 7 d. MIC titres from ≥4 independent assays are reported as geomean ± SD and group titres as Mean ± SEM.

### 4.2. Time-Kill Assay

#### 4.2.1. Vegetative Cells

Time kill studies were determined in Sabouraud Liquid Medium (SLM) and distilled water (DW) for *C. albicans* and in DW for *A. fumigatus* hyphae. Overnight cultures in Sabouraud Liquid Medium (SLM, Oxoid, Cambridge, UK) were diluted 1/100 and incubated 2 h at 37 °C on a platform shaker at 160 rpm and log-phase suspensions prepared in SLM or DW with 1% dimethyl sulfoxide (DMSO) to give final densities of 1–5 × 10^6^ blastospores (*C. albicans*) and ~1 × 10^5^ hyphal fragments (*A. fumigatus*). The MICi for *C. albicans* and *A. fumigatus* was determined as above. NPBD 128–512 mg/L in SLM and DW and SLM controls, were inoculated and incubated with shaking at 37 °C or room temperature (RT) for *C. albicans* and RT for *A. fumigatus*. At intervals from 0–8 h for *C. albicans* and 0–48 h for *A. fumigatus*, 0.1 mL aliquots were serially diluted (log_10_) in 0.85% saline and 0.01 mL aliquots spread-plated to Sabouraud Dextrose Agar (SDA, Oxoid), incubated to 48 h at 37 °C for *C. albicans* and to 96 h at 30 °C for *A. fumigatus* and colony counts recorded. The kill rate is reported as reduction in log_10_ viable counts over time (kill curves) and as log_10_ reduction factors (RF) at time (T) determined as log_10_ count untreated–log_10_ count treated.

#### 4.2.2. Microconidia

Microconidia from a culture of *A. fumigatus* (25 mL SDA, tissue culture flask, 5 d, RT) were harvested by glass beads in 10 mL saline/T20 abrasion of colony surface and the suspension heated at 56 °C 30 min to kill hyphae and spore density determined by haemocytometer count. 1 mL volumes of DW/1% T20 with 0 or 512 mg/L NPBD at RT (~20 °C) were inoculated to give a final inoculum density 3–5 × 10^5^ microconidia. Log_10_ dilutions (saline/1% T20) of aliquots taken at 0, 2, 6 and 24 h were spread onto SDA, incubated 5 d at RT and geomean counts from 3 assays reported.

### 4.3. In Vitro Resistance MIC Assay

NPBD dilution sets (256 mg/L–0.25 mg/L and 1% DMSO) were prepared in SLM. Set 1 for each strain was inoculated with an overnight SLM culture prepared as above to give final densities of 5 × 10^3^–1 × 10^4^ cfu/mL, and incubated at 25 °C, the initial strain MIC_W0_ recorded at 2 d, and set re-incubated to 7 d and the 7 d set MIC_W1_ recorded as the lowest concentration inhibiting growth. The highest dilution showing turbidity was diluted to (≤1 × 10^5^ cfu/mL) inoculated into W2 set and at 7 d MIC_W2_ recorded. This procedure was repeated to Week 12 for each strains. A 0.01 mL sample from W_12_ was spread-plated to SDA and selected colonies serially passaged 3 times in SLM. The strain MIC_W13_ for colonies was determined as for W_0_. Acquisition of resistance to NPBD was defined as ≥8-fold increase in MIC_W13_ compared to MIC_W0_.

## 5. Patents


Denisenko, P.P.; Sapronov, N.S.; Tarasenko, A.A. Antimicrobial and radioprotective compounds. US Pat 9,045,452, 2 June 2015. *[Claim: A method for the treatment of a gastrointestinal infection]*Denisenko, P.P.; Sapronov, N.S.; Tarasenko, A.A. Antimicrobial and radioprotective compounds. US Pat 8,569,363, 29 October 2013. *[Claim: A method for the therapeutic treatment of a skin or soft tissue infection]*Denisenko, P.P.; Sapronov, N.S.; Tarasenko, A.A. Antimicrobial and radioprotective compounds. US Pat 7,825,145, 2 November 2010. *[Claim: A method of treating vulvo-vaginitis]*Nicoletti, A.; White, K. Protein tyrosine phosphatase modulators. WO/2008/061308, 29 May 2008.


## Figures and Tables

**Figure 1 antibiotics-11-01188-f001:**
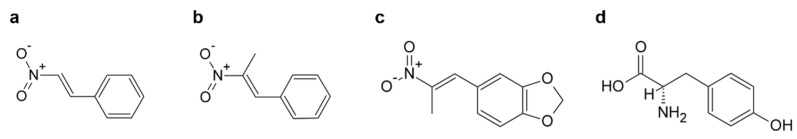
Structures of the active pharmacophores of the nitroalkenyl benzenes: (**a**) NEB ([(E)-2-nitroethenyl]benzene) and (**b**) NPB ((2-nitroprop-1-enylbenzene) and (**c**) NPBD ((5-[(E)-2-nitroprop-1-enyl]-1,3-benzodioxole) an active antimicrobial mimetic of (**d**) tyrosine (2-amino-3-(4-hydroxyphenyl)propanoic acid).

**Figure 2 antibiotics-11-01188-f002:**
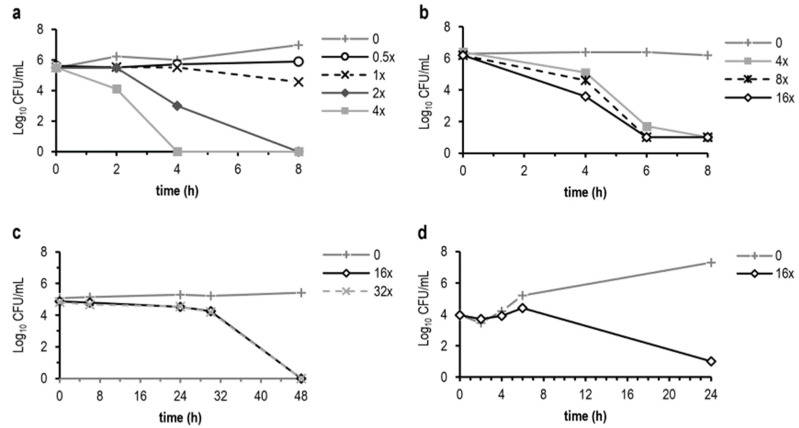
Representative kill curves at multiples of the MICi for NPBD against (**a**) *Candida albicans* (ATCC 10231) in SLM at 37 °C, (**b**) *C. albicans* in distilled water at room temperature, (**c**) *Aspergillus fumigatus* (RMIT 702/1-1) hyphae and (**d**) microconidia in distilled water at room temperature.

**Figure 3 antibiotics-11-01188-f003:**
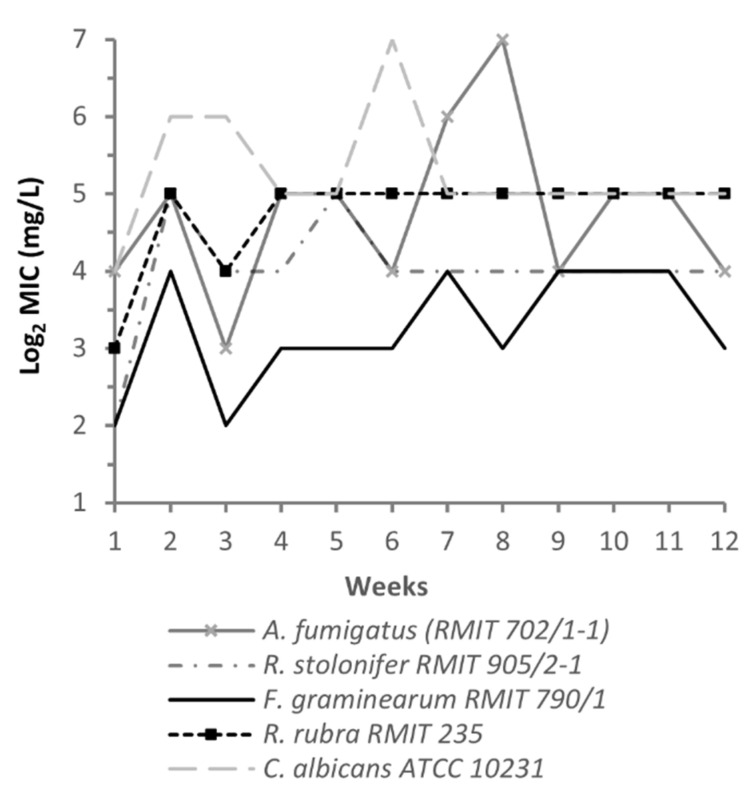
Resistance to NPBD in fungal strains on 12-week exposure. Test species were inoculated into NPBD 2-fold dilution sets (0.25 to 256 mg/L + 1% DMSO in SLM) (W_0_), incubated at 25 °C for 2 d then to 7 d, and subcultured weekly into fresh dilution sets to obtain 7-day MIC (MIC_W1_ to MIC_W12_). Colonies from W_13_ were serially subcultured 3 times in drug free medium to end of W_13_.

**Table 1 antibiotics-11-01188-t001:** Antifungal activity (MIC and MFC) ^a^ of NPBD, Amphotericin B and Miconazole.

PHYLUM		NPBD		Amphotericin B		Miconazole	
Fungal group description							
**Order, Family** ^b^							
	Species	MIC	MFC	MIC	MFC	MIC	MFC
**ASCOMYCOTA**							
Saprophytic filamentous species							
**Eurotiales, Aspergillaceae**							
	*Aspergillus flavus* RMIT 312	16	32	8	8	ND	ND
	*A. niger* RMIT 582	8	16	2	2	ND	ND
	*A. fumigatus* RMIT 702/1-1	6.2 ± 4	16 ± 12	1	8	0.3	2.0
	*Penicillium chrysogenum*	1.3 ± 0.6	2.0	ND	ND	ND	ND
**Eurotiales, Thermoascaceae**							
	*Paecilomyces variotii* ^c^	1	2	1	1	ND	ND
**Hypocreales, Nectriaceae**							
	*Fusarium graminearum* RMIT 790/1	2.8 ± 1	5.2 ± 2	5.2 ± 2	12 ± 2	1	8
	*Fusarium chlamydosporum* RMIT 603	8	8	2	8		
Dimorphic filamentous species							
**Chaetothyriales, Herpotrichiellaceae**							
	*Fonsecaea pedrosoi*	2.8 ± 3	3.4 ± 7	5.7 ± 3	9.5 ± 7	ND	ND
	*Phialophora verrucosa* RMIT 142	16	32	2	4	ND	ND
**Capnodiales, Teratosphaeriaceae**							
	*Hortaea werneckii* RMIT 115 (*Cladosporium werneckii*)	8	8	ND	ND	4	4
**Microascales, Microascaceae**							
	*Scedosporium apiospermum* (*S. boydii*) RMIT 141	2	4	4	>16 ^d^	8	8
**MUCOROMYCOTA/ZYGOMYCOTA**							
Saprophytic filamentous species							
**Mucorales, Rhizopodaceae**							
	*Rhizopus stolonifer* RMIT 905/2-1	4	11 ± 5	>16 ^d^	>16 ^d^	8	16
	*Rhizopus oryzae*	64	64	4	4	ND	ND
**Mucorales, Lichtheimiaceae**							
	*Rhizomucor pusillis*	4	8	1	1	ND	ND
**Filamentous species (*n* = 14)** **Average ± SD** ^d^		**10.2 ± 16.2**	**15.1 ± 17.2**	**5.7 ± 8.6**	**10.1 ± 10.8**	**4.3 ± 3.7**	**7.6 ± 5.4**
**ASCOMYCOTA**							
Parasitic dermatophytic species							
**Onygenales, Arthrodermataceae**							
	*Epidermophyton floccosum*	0.5	16	ND	ND	0.25	0.25
	*Trichophyton rubrum* S1	1.4 ± 5	256	1	256	1.7 ± 0.5	64
	*Trichophyton rubrum* S2	1	128	4	16	1	8
	*Microsporum canis*	0.7 ± 0.7	13.5 ± 4	0.5	16	ND	ND
	*Microsporum gypseum*	1.3 ± 0.5	9.2 ± 3.6	1	8	5.7 ± 2.3	≥64
**Saccharomycetales, Debaryomyceaceae**							
	*Candida albicans* ATCC 10231	9 ± 4	9.5 ± 3.6	0.25	2	0.5	0.5
	*Candida glabrata* RMIT 157	2 ± 1	2	0.5	1	0.25	0.5
	*Candida guilliermondii* (*Pichia*) RMIT 176	2.4 ± 1	2.4 ± 1	0.03	0.06	1.2 ± 0.5	1.2 ± 0.5
	*Candida krusei* (*Pichia kudriavzeuii*) anamorph RMIT 177	3.4 ± 1	3.4 ± 1	0.25 ± 0.2	0.3 ± 0.2	1.7 ± 0.5	1.7 ± 0.5
	*Candida parapsilopsis* RMIT 178	2.4 ± 1	2.4 ± 1	0.3 ± 0.14	0.4 ± 0.35	0.7 ± 0.75	0.7 ± 0.75
	*Candida tropicalis* RMIT 181	4	4	0.25	0.4 ± 0.14	0.8 ± 0.3	1.3 ± 0.8
**BASIDIOMYCOTA**							
Saprophytic yeasts							
**Tremellales, Cryptococcaceae**							
	*Cryptococcus neoformans*	1	1	0.5	0.5	2.8 ± 1.1	4
**Sporidiobolales, Sporidiobolaceae**							
	*Rhodotorula rubra* (*mucilaginosa*)	9.2 ± 3.6	9.2 ± 3.6	ND	ND	8	18.4 ± 7
	**‘Yeasts’ Average (*n* = 8) ± SD**	**3.9 ± 1.5**	**4.2 ± 3.3**	**0.3 ± 0.2**	**0.7 ± 0.7**	**2.0 ± 2.6**	**3.5 ± 6.1**
	**All Fungi (*n* = 27)-Average ± SD**	**7.0 ± 1.8**	**25.6 ± 2.2**	**3.3 ± 0.16**	**18.4 ± 0.22**	**2.7 ± 0.8**	**11.9 ± 0.59**

^a^ Broth microdilution assays were based on CLSI methods for yeast and filamentous fungi [58,59]. MIC (100%) and MFC (>99.9%) at 48 h, 72 h or 7 d, depending on growth rate assessed by turbidity of growth control. Average of ≥4 independent replicates ± SD. ND, not determined. SD = 0 where not reported. The QC strain of *C. albicans* ATCC 10,231 for AMB was within the prescribed range. ^b^ NCBI phylogenetic classification of fungi. Strains not identified by ATCC or RMIT collection number are environmental or clinical isolates. ^c^ 7 clinical isolates of *C. albicans* were tested against NPBD (mean MIC 12.5 (±15.2), MFC 15.2 (±7.3) mg/L) and MCZ (mean MIC 2, MFC 4 (±0.7) mg/L). ^d^ For AMB where MIC/MFC reported as greater than the highest tested concentration, the 2-fold higher value was used to estimate the mean.

**Table 2 antibiotics-11-01188-t002:** Comparison of the activity ^a^ of NPBD, AMB and MCZ for major fungal groups.

Test Compound	NPBD			AMB			MCZ		
Fungal Group ^b^ (*n*)	MIC	MFC	MFC/MIC	MIC	MFC	MFC/MIC	MIC	MFC	MFC/MIC
All Saprophytic opportunistic filamentous species (13) ^c^	11.4	16.4	1.4	6.2	8.4	1.4	3.1	8.7	2.8
Endemic dimorphic species (4)	7.2	11.9	1.7	3.9	15.2	3.9	6	6	1
Parasitic dermatophytic species (5)	1	84.5	84.5	1.6	74	46.3	2.2	34.1	15.5
Saprophytic/commensal *Candida* species (6)	3.9	4	1	0.3	0.7	2.3	0.9	1	1.1

^a^ Broth microdilution assays (were based on CLSI methods for yeast and filamentous fungi [58,59]. MIC (100%) and MFC (>99.9%) at 48 h, 72 h or 7 d, depending on growth rate assessed by turbidity of growth control. Average of ≥4 independent replicates per species. The QC strain of *C. albicans* ATCC 10231 for AMB was within the prescribed range. ^b^ Groupings of fungi selected from Table 1 are based on major clinical infection types. The basidiomycete yeasts *C. neoformans* and *R. mucilaginosa* were excluded as only one representative from each pathogen type was tested. ^c^
*P. chrysogenum* was excluded from filamentous species for lack of AMB and MCZ titres.

## Data Availability

Data is contained within the article or Appendix A.

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
