# Peer review of "The Anti-Fungal Activity of Nitropropenyl Benzodioxole (NPBD), a Redox-Thiol Oxidant and Tyrosine Phosphatase Inhibitor"

_antibiotics, 2022, doi:10.3390/antibiotics11091188_

Round 1
Reviewer 1 Report
The manuscript reports the phenotypic effects of NPBD on saprophytic, commensal, and phylogenetically diverse parasite species. In addition, NPBD was briefly evaluated concerning Amphotericin B and Miconazole, agents with a similar activity spectrum and metabolic effects beyond their main targets. It contains some very interesting results and is suitable for publication after revisions.
Major points:
Page 4, Lines 169-170: This information must be described in Materials and Methods "27 sap- 169 trophytic, commensal and parasitic species from 3 orders and 12 families"
Page 4, Table 1: Table 1 must be reviewed for presentation, format, captions, and footnotes.
Why did the authors not determine the antifungal activity of NPBD, Amphotericin B, and Miconazole for all 27 saprophytic, commensal and parasitic species?
The table must indicate what was not done and not have empty spaces
Page 6, Line 185: Replace Scedosporium apiospermum with S. apiospermum
Authors should fully describe the meaning of MIC and MFB the first time they appear in the text
Page 6, Table 2: Table 2 must be reviewed for presentation, format, captions, and footnotes.
This table shows the results of 25 fungal groups, not of the 27 included in the study.
Page 6, Line 196: Authors must fully describe the full name of P. murina the first time they appear in the text.
Page 6, Line 202: Put C. albicans in italic
Page 7, Line 252 and 253: Replace Candida albicans and Cryptococcus neoformans with C. albicans and C. neoformans
Page 9: Where is the text's indication of Figure 2b?
Page 9, Line 306: The font size is not correct
Page 9, Line 332: Describe the graph d of figure 2
Page 9, Line 347: Correct 2d
Must improve the quality of Figure 3.
Page 11, Line 411: Authors must fully describe the meaning of ROS the first time they appear in the text.
Page 13, Line 474: Authors should describe in this section which strains were included in the study and with the full name (A. flavus, A. fumigatus, S. apiospermum, T. rubrum, and C. parapsilosis)
Page 13, Line 486: Authors should fully describe the meaning of SLM and DW the first time they appear in the text.
Page 13, Line 489: Authors must fully describe the meaning of DMSO the first time they appear in the text.
Page 13, Line 492: Authors must fully describe the meaning of RT the first time they appear in the text.
Page 13, Line 515: Endpoint placement after strains
Page 13, Line 516: Authors must fully describe the meaning of SDA the first time they appear in the text
Page 14-21, 549-860: References, uniformize low cases or capital letters.
Reviewer 2 Report
The manuscript entitled: The anti-fungal activity of a nitroalkenyl benzene derivative, a
redox-thiol oxidant and tyrosine phosphatase inhibitor by the authors Gina Nicolettiet al., have done a great job by determining the anti-fungal activity of a nitroalkenyl benzene derivative .There are some flaws that shall be addressed .some of them are:
1.The title of the manuscript must be changed as:
The anti-fungal activity of a nitroalkenyl benzene derivative: A
redox-thiol oxidant and tyrosine phosphatase inhibitor.
2. Add some lines for the determination of MIC in the methodology section at the end.
3. Add conclusion of the manuscript at the end.
4. Yet I am not a native speaker of English language but still I recommend that the English language needs touching up in a major way. The article needs to be rewritten in readable English. Many sentences are confusing, do not lead to scientific meaning, and can be found starting in lower case, and upper case can be detected in the middle of sentences without proper nouns.
Round 2
Reviewer 1 Report
The suggested changes were incorporated into the manuscript and as we have no further suggestions, we consider it ready for publication.